# Clinical and Laboratory Findings of Nosocomial Sepsis in Extremely Low Birth Weight Infants According to Causative Organisms

**DOI:** 10.3390/jcm11010260

**Published:** 2022-01-04

**Authors:** Kyung-Hee Park, Su-Jung Park, Mi-Hye Bae, Seong-Hee Jeong, Mun-Hui Jeong, Narae Lee, Young-Mi Han, Shin-Yun Byun

**Affiliations:** 1Department of Pediatrics, Pusan National University Hospital, Pusan National University School of Medicine, Busan 49241, Korea; pnuhpkh@gmail.com (K.-H.P.); psj0430@gmail.com (S.-J.P.); piglet314@hanmail.net (M.-H.B.); 2Department of Pediatrics, Pusan National University Children’s Hospital, Pusan National University School of Medicine, Yangsan 50612, Korea; ibory830@naver.com (S.-H.J.); maldives8@hanmail.net (M.-H.J.); whitecloud11@hanmail.net (N.L.); skybluehym@gmail.com (Y.-M.H.)

**Keywords:** sepsis, preterm infant, extremely low birth weight

## Abstract

Background: nosocomial sepsis remains a significant source of morbidity and mortality in extremely low birth weight (ELBW) infants. Early and accurate diagnosis is very important, but it is difficult due to the similarities in clinical manifestation between the causative microorganisms. We tried to identify the differences between causative microorganisms in clinical and laboratory findings and to help choose antibiotics, when sepsis was suspected in ELBW infants. Methods: a retrospective study was conducted on preterm infants, born at less than 28 weeks of gestation, with a birth weight of less than 1000 g between January 2009 and December 2019. Clinical and laboratory findings of suspected sepsis, after the first 72 h of life, were assessed. We classified them into four groups according to blood culture results (gram positive, gram negative, fungal, and negative culture groups) and compared them. Results: a total of 158 patients were included after using the exclusion criteria, with 45 (29%) in the gram positive group, 35 (22%) in the gram negative group, 27 (17%) in the fungal group, and 51 (32%) in the negative culture group. There were no significant differences in mean gestational age, birth weight, and neonatal morbidities, except for the age of onset, which was earlier in the fungal group than other groups. White blood cell (WBC) counts were the highest in the gram negative group and the lowest in the fungal group. The mean platelet counts were the lowest in the fungal group. C-reactive protein (CRP) levels were the highest in the gram negative group, while glucose was the highest in the fungal group. Conclusions: in conclusion, we showed that there are some differences in laboratory findings, according to causative microorganisms in the nosocomial sepsis of ELBW infants. Increased WBC and CRP were associated with gram negative infection, while decreased platelet and glucose level were associated with fungal infection. These data may be helpful for choosing empirical antibiotics when sepsis is suspected.

## 1. Introduction

Nosocomial sepsis is generally defined as a bloodstream infection that presents after the first 72 h of life among infants hospitalized in the neonatal intensive care unit (NICU). Extreme prematurity is one of the greatest risk factors for nosocomial sepsis. More than 65% of ELBW infants are treated for clinical, or proven, neonatal infection during their hospitalization, and the associated mortality is 20–40% [1,2].

There are multiple factors that put extremely low birth weight (ELBW) infants at particularly high risk, such as prolonged hospitalization, central venous catheters and parenteral nutrition, endotracheal intubation and mechanical ventilation, and lack of enteral feeding [3]. Due to the high risk of nosocomial infection and the associated morbidity in ELBW infants, these patients are frequently exposed to broad-spectrum antibiotics [4].

Epidemiological data on neonates show that the predominant pathogens causing nosocomial sepsis are coagulase negative staphylococci (CoNS), followed by gram-negative bacteria and fungi. Because CoNS is usually resistant to a number of commonly used antibiotics, neonatologists usually add vancomycin to the empirical combination when treating suspected nosocomial sepsis in ELBW infants [4]. However, early and accurate diagnosis of nosocomial sepsis is challenging because the presenting signs and symptoms are subtle and nonspecific [5]. In addition, invasive bacterial and fungal infections are all similar in clinical manifestation [6].

In order to diagnose nosocomial sepsis early and choose the right empirical antibiotics while avoiding broad-spectrum antibiotics or vancomycin, we tried to identify the differences between causative microorganisms, using clinical findings and routine laboratory findings, when sepsis was suspected in ELBW infants.

## 2. Materials and Methods

### 2.1. Materials

A retrospective study was conducted on preterm infants, born at less than 28 weeks of gestation, with a birth weight of less than 1000 g between January 2009 and December 2019 at Pusan National University Yangsan Hospital.

Neonates with suspected nosocomial sepsis were defined as having one or more of the following signs: apnea, acutely ill appearance, decreased activity, fever (>37.8 °C), lethargy, irritability, gastrointestinal dysfunction with milk intolerance, hypotension, and sudden increased respiratory support after the first 72 h of life.

Patients were included if their laboratory tests, including blood culture, were taken as these symptoms and signs started. For the purpose of our investigation, episodes of polymicrobial sepsis were excluded, and in infants with multiple episodes of nosocomial sepsis, only the first episode was included.

We also excluded all infants with other abnormalities, including congenital malformation, or chromosomal abnormalities and congenital infections, such as cytomegalo virus.

We classified them into four groups according to the results of blood culture: the gram positive, gram negative, fungal, and negative culture groups.

### 2.2. Methods

The data, including clinical characteristics and morbidities, were reviewed retrospectively.

The perinatal data were collected from the medical charts of the infants and included the parameters of gender, gestational age (GA), and birth weight (BW).

The following neonatal characteristics were reviewed: respiratory distress syndrome (RDS); patent ductus arteriosus (PDA; defined as needing either medical therapy with ibuprofen or surgical ligation); intraventricular hemorrhage (IVH; defined as grade ≥III); bronchopulmonary dysplasia (BPD; defined as moderate and above); periventricular leukomalacia (PVL, defined as grade 2 and above); retinopathy of prematurity (ROP, defined as stage 2 and above)

### 2.3. Statistical Analysis

Values of all non-normally distributed variables are expressed as medians and interquartile ranges (IQR; 25–75%). For comparing the categorical data of the groups, the chi-square test was used. A Kruskal–Wallis test and post-hoc analysis were performed for pairwise comparison of subgroups to verify differences across the patient groups. The statistical analyses were performed using MedCalc software (version 16.4.3; MedCalc, Mariakerke, Belgium) and RStudio (version 1.2b; RStudio, Boston, MA, USA). Receiver operating characteristics (ROC) and area under the curve (AUC) were generated to determine the optimal cut-off value for each marker in detecting sepsis, and to calculate sensitivity, specificity, positive predictive value (PPV), and negative predictive value (NPV). A *p*-value of <0.05 was considered significant.

## 3. Results

During the study period, 455 preterm infants, born at less than 28 weeks of gestation, with a BW of less than 1000 g, were admitted at our hospital. A total of 161 patients underwent laboratory tests for suspected sepsis after the first 72 h of life. Of these, three were excluded for polymicrobial sepsis (*n* = 1), congenital CMV infection (*n* = 1), and chromosomal abnormality (*n* = 1). Finally, 158 patients were included, with 45 (29%) in the gram positive group, 35 (22%) in the gram negative group, 27 (17%) in the fungal group, and 51 (32%) in the negative culture group. A detailed flowchart of the study population is shown in Figure 1. The predominant organisms cultured in culture positive groups were *Staphylococcus aureus*, *Klebsiella pneumonia*, and *Candida parapsillosis*, respectively.

The demographic and clinical characteristics of the four groups are shown in Table 1. The mean GA was 25.5 (25.0–26.5) weeks, 25.4 (24.3–26.4) weeks, 25.3 (24.3–26.2), and 25.9 (24.3–26.3) weeks in the gram positive, gram negative, fungal, and negative culture groups, respectively. The mean BW was 790 (710–910) g, 790 (685–875) g, 760 (685–835) g, and 790 (705–920) in the same sequence, respectively. There were no significant differences in mean GA and BW. No significant differences were observed between the four groups in terms of neonatal morbidities including RDS, PDA, and BPD. However, the age of onset was lower in the fungal group than in the other groups (34, 31, 21, and 29 days of life in the gram positive, gram negative, fungal, and negative culture groups, respectively; *p* = 0.001). Mortality rate was highest in the fungal group (28.9%, 54.3%, 66.7%, and 19.6% in the gram positive, gram negative, fungal and negative culture groups, respectively; *p* = 0.001).

White blood cell (WBC) counts were the highest in the gram negative group and the lowest in the fungal group, while remaining normal in the gram positive group and negative culture group. The mean platelet counts were the lowest in fungal group (137,000/mm^3^, 100,000/mm^3^, 61,000/mm^3^, and 199,000/mm^3^ in the gram positive, gram negative, fungal, and negative culture groups, respectively; *p* < 0.001). C-reactive protein (CRP) levels were the highest in the gram negative group (2.2 mg/L, 6.8 mg/L, 1.13 mg/L, and 2.29 mg/L, respectively, in each group (*p* < 0.001)). Glucose levels were the highest in the fungal group, while the other three groups showed normal ranges (133 mg/dL, 130 mg/dL, 253 mg/dL, and 97 mg/dL in the gram positive, gram negative, fungal, and negative culture groups, respectively). (Table 2)

The optimal cut-off values for CRP, platelets, and glucose were identified by drawing ROC curves. For gram negative infection, the cut-off value for the CRP was found to be 4.5 mg/dL, with sensitivity, specificity, PPV, and NPV of 82.9, 78.0, 51.8, and 94.1%, respectively. The AUC for CRP was 0.85 (95%CI: 0.791–0.91, *p* < 0.001). For fungal sepsis, the optimal cut-off value for platelet counts was 89,000/µL, with sensitivity, specificity, PPV, and NPV of 100.0, 70.2, 40.9, and 100.0%, respectively. The AUC for platelet was 0.829 (95%CI: 0.767–0.892, *p* < 0.001). The optimal cut-off value for glucose level was 182.0 mg/dl, with 96.3% sensitivity, 88.5% specificity, 63.4% PPV, and 99.1% NPV for diagnosing fungal infections. The AUC for glucose was 0.954 (95%CI: 0.923–0.984, *p* < 0.001) (Figure 2).

## 4. Discussion

The clinical presentation of nosocomial sepsis in ELBW infants is subtle, and as invasive bacterial and fungal infections are similar, this may cause diagnostic delay [6]. In this study, we were unable to identify differences in clinical findings between microorganisms, except in the age of onset, which was the lowest in the fungal group. At birth, most infants are uncolonized or have low colony counts of yeast. In the absence of antifungal prophylaxis, up to 60% of ELBW infants become colonized in the first 2–3 weeks after birth [7]. Clerihew et al. described a median age at diagnosis of 14 days, using 94 cases of invasive fungal infection in ELBW infants [8]. According to our study, the mean age of onset was 21 days after birth. These results are consistent with the fungal colonization timings mentioned above. Therefore, it is speculated that the fungal infection may occur as soon as colonization occurs in ELBW infants with compromised immunity. Our study was designed to include the first episode of nosocomial sepsis in order to avoid the confounding effects of previous infections. Some patients, with fungal sepsis as second or third episodes, were excluded in this study, as it has been shown that fungal sepsis tends to occur in a setting of prolonged broad-spectrum antibiotics after first gram positive or negative sepsis. Therefore, our study was not able to explain the exact prevalence and onset time of each organism. However, Guida et al. reported that the fungal sepsis had 8% of prevalence and the earliest onset age, when compared with gram positive or gram negative sepsis, in very low birth weight (VLBW) infants [9]. That study was consistent with our findings.

We found that the mortality rate was highest in the fungal group. Greenberg et al. reported that the mortality rate of nosocomial sepsis of ELBW infants were 15%, 18%, and 39% in gram positive, gram negative, and fungal group, respectively [10]. In that study, the mortality rate was highest in the fungal group, as with our results, although they were not statistically significant. There were no differences of morbidities according to the microorganisms in our study, so the explanation for this finding is unclear and needed further studies.

Although blood culture results are important for diagnosing neonatal sepsis, its unavailability within the 2–3 days of incubation and low culture rate make early diagnosis difficult. For the rapid identification of microorganisms causing sepsis, novel laboratory methods, such as cytokine and molecular analyses, have been developed; however, it is unlikely that these methods will be useful in the near future because they are not very cost effective [11,12]. So far, reliable laboratory diagnosis has not been achieved.

The complete blood cell (CBC) count is a rapid, inexpensive, and widely available diagnostic test [13,14]. However, the diagnostic accuracy of the CBC count is not well-defined in neonatal sepsis, and the usefulness of the CBC count has reported conflicting results in preterm infants, especially in VLBW infants [15,16]. Hornik et al. studied the diagnostic accuracy of CBC count, in nosocomial sepsis in a large multicenter population, and reported CBC parameters associated with nosocomial sepsis, including a total WBC count of <5000/mm^3^ and an immature neutrophil/total neutrophil (I/T) ratio of >0.10 [17]. According to some studies on the pathogenesis of organisms responsible for sepsis, elevated I/T ratios were significantly more common in gram negative sepsis than in gram positive ones [18]. Dogan et al. analyzed red-cell distribution width (RDW) during a sepsis episode and its association with the type of growing microorganism [19]. According to that study, RDW levels increased in preterm infants with sepsis, which was especially evident in gram negative infections. Unfortunately, we did not check I/T ratio and RDW levels in this study. Nevertheless, we found that higher WBC counts were associated with gram negative sepsis in ELBW infants. However, CBC count varies significantly with day of life and GA, and it is affected by non-infectious comorbidities. More specifically, VLBW infants are known to develop physiologic late onset neutropenia, which does not correlate with sepsis. Therefore, we have to cautiously interpret the WBC count during the diagnosis of sepsis in preterm infants.

Thrombocytopenia has been used as an early, but nonspecific, marker for sepsis [9]. Several studies have shown quantitative differences in the platelet response to infection, with the three major categories of organisms causing sepsis in the VLBW infants. Daniel et al. reported that thrombocytopenia, at the time of blood culture, was a risk factor for candidemia [20]. In our study, mean platelet count was 54,000/mm^3^ in the Fungal group, which was significantly lower than those of the other three groups. This result was similar to previously published results. According to Benjamin et al., fungal sepsis is associated with a greater degree of thrombocytopenia than CoNS sepsis [6]. Although the mechanisms for this result have not yet been identified, it is known that administration of a platelet activating factor is protective in a mouse model of *C. albicans* sepsis, suggesting that platelet activation plays a role in the host defense against fungal pathogens [21]. On the other hand, Ree et al. demonstrated evidence of a relationship between gram negative infections and thrombocytopenia [22]. Further work is needed to understand the basis for the effects of different species of microorganisms.

The most extensively studied laboratory marker for determining the diagnosis of neonatal sepsis has been CRP [23]. The increased survival of extremely premature infants has brought the diagnostic validity of the method to the point of great scientific controversy. Some clinicians suggest that CRP is an inappropriate method in extremely premature infants, as they are incapable of sufficiently increasing CRP levels following a septic event. One possible explanation for this is that an extremely immature liver cannot respond to septic stimuli to the same extent as those in more mature premature and full term infants [24,25].

On the other hand, in a study of 123 ELBW infants, with a mean GA of 27 weeks and mean BW of 1000 g, Wagle et al. concluded that infants with gram negative sepsis are capable of mounting significant CRP responses similar to those of full term infants [26]. In addition, Dritsakou et al. found that gram negative microorganisms were associated with higher CRP levels in ELBW infants [25]. On the contrary, some studies found that CRP could not serve to disclose CoNS infection. Previous studies have suggested that CoNS are associated with lower levels of inflammation than other bacteria [27,28,29]. These studies showed that CoNS, involving *S. epidermidis*, does not cause local inflammatory reactions or immunoglobulin G responses in animal models [29].

The results of our study are consistent with those of the abovementioned studies to some extent. According to our study, CRP increased in the gram negative group, whereas CRP in the gram positive group did not increase as much. However, in our study, the majority of the gram positive group had *S. aureus* involvement, and not CoNS. CoNS are a common skin commensal and the most frequent organism associated with nosocomial infection in ELBW infants. However, recent studies have suggested that CoNS infection has decreased with central intravascular line care protocols and the reduction in the use and duration of central lines, which is consistent with our result [30,31]. Therefore, we do not know exactly why the gram positive group showed lower CRP level than the gram negative group.

It has been recognized that hyperglycemia may be an important early sign in neonatal sepsis [32,33]. It is known that inadequate hepatic and diminished pancreatic insulin secretory responsiveness increase the risk of hyperglycemia in stressful episodes, such as sepsis, in preterm infants [34]. Interestingly, blood glucose was the highest in the fungal group than in the other groups in our study. This result is consistent with another study by Manzoni et al., which revealed that hyperglycemia is more frequent in fungal sepsis than in bacterial sepsis in preterm infants [33,35]. They suggested that hyperglycemia could be a possible surrogate marker predictor of invasive fungal infection in preterm infants. However, Levit et al. reported that sepsis-related death was associated with hypoglycemia in ELBW infants, and there was no specific correlation between glucose disturbances and the type of pathogen [36]. Because these reports had different time of tests, including not only initial blood sampling but also nadir values, caution is needed in the interpretation, and further studies are needed to know this association.

This study has several limitations. First of all, we assessed patients with suspected sepsis from medical records due to retrospective study design. Therefore, there might be a selection bias in patients’ collection. The sample size is too small to identify the differences between three groups. Further well-designed prospective studies with larger case numbers are needed.

## 5. Conclusions

In conclusion, we have shown that there are some differences in laboratory findings, according to causative microorganisms in nosocomial sepsis of ELBW infants. Increased WBC and CRP were associated with gram negative infection, while decreased platelet and glucose levels were associated with fungal infection. These data may be helpful to choose empirical antibiotics when sepsis is suspected, as well as important to reduce the overuse and potential for bacterial resistance to broad spectrum antibiotics.

## Figures and Tables

**Figure 1 jcm-11-00260-f001:**
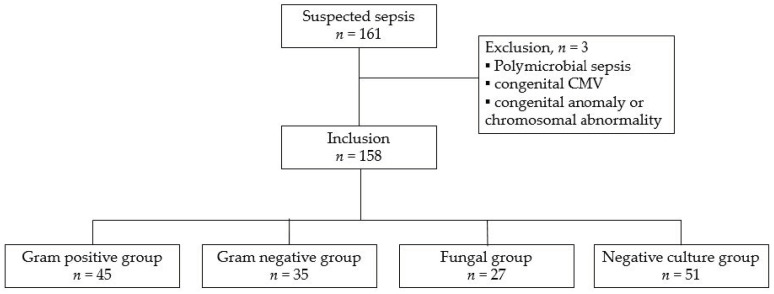
Study selection flowchart.

**Figure 2 jcm-11-00260-f002:**
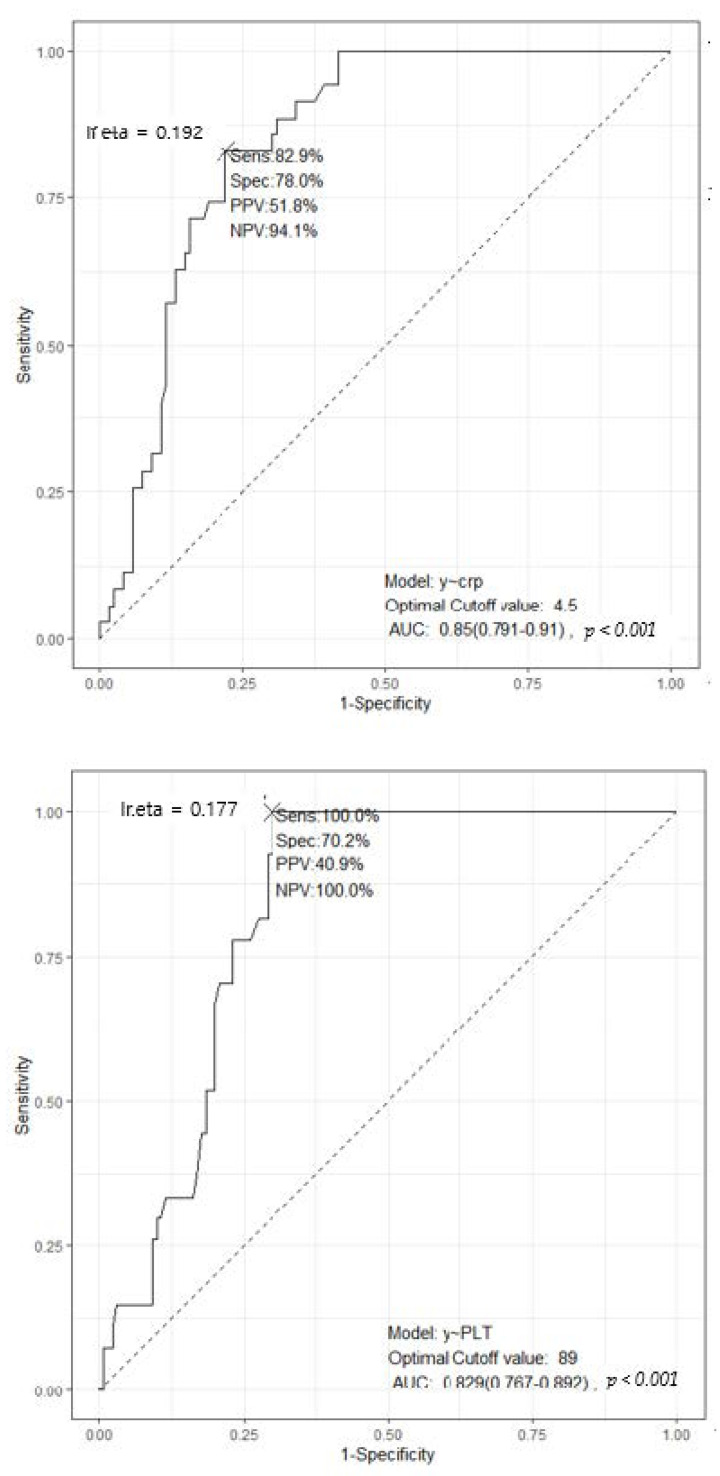
Receiver operating characteristics curves, calculated for cut-off value of CRP, platelet, and glucose.

**Table 1 jcm-11-00260-t001:** Comparison of demographic and clinical characteristics, according to microorganisms.

	Gram Positive	Gram Negative	Fungal	Negative Culture	*p* Value
Gestational age	25.5 (25.0–26.5)	25.4 (24.5–26.4)	25.3(24.3–26.2)	25.9 (24.3–26.3)	0.415
Birth weight	780 (695–900)	790 (710–910)	760 (685–835)	790 (705–920)	0.580
Male gender	27 (60%)	19 (54%)	16 (59%)	25 (49%)	0.865
Inborn	41 (91%)	28 (80%)	21 (78%)	39 (77%)	0.234
Onset (age, d)	34 (26–61)	31 (19–59)	21 (11–31)	29 (20–55)	0.001 *
RDS	45 (100.0%)	35 (100.0%)	27 (100.0%)	50 (98.0%)	0.550
PDA	16 (35.6%)	19 (54.3%)	14 (51.9%)	19 (37.3%)	0.223
IVH	6 (13. 3%)	6 (17.1%)	9 (33.0%)	7 (13.7%)	0.106
BPD	42 (93.3%)	28 (80.0%)	23 (85.2%)	43 (84.3%)	0.205
ROP	7 (15.6%)	4 (11.4%)	0 (0.0%)	8 (15.7%)	0.181
PVL	4 (8.9%)	2 (5.7%)	1 (3.7%)	5 (9.8%)	0.748
Mortality	13 (28.9%)	29 (54.3%)	18 (66.7%)	10 (19.6%)	<0.001 *

Mean (range) for gestational age, birth weight, and onset; * *p* value < 0.05, RDS Respiratory distress syndrome; PDA Patent ductus arteriosus; IVH Intraventricular hemorrhage; BPD Bronchopulmonary dysplasia; ROP Retinopathy of prematurity; PVL Periventricular leukomalasia.

**Table 2 jcm-11-00260-t002:** Comparison of main laboratory findings, according to microorganisms.

	Gram Positive	Gram Negative	Fungal	Negative Culture	*p* Value
WBC	11440 (7220–18740)	16880 (9005–26005)	9080 (5585–13325)	11940 (7425–17415)	0.044 *
Hb	10.9 (9.7–11.9)	10.6 (9.0–12.4)	10.9 (9.8–12.0)	11.3 (10.5–12.5)	0.305
PLT	137,000 (83,000–225,000)	100,000 (48,000–185,500)	61,000 (32,000–70,000)	199,000 (106,500–271,000)	<0.001 *
CRP	2.2 (1.6–4.6)	6.8 (4.7–11.6)	1.13 (0.4–3.3)	2.29 (0.96–4.05)	<0.001 *
Glucose	133 (99–155)	130 (102–167)	253 (219–288)	97 (79–131)	<0.001 *
ph	7.34 (7.27–7.37)	7.29 (7.21–7.38)	7.28 (7.18–7.34)	7.31 (7.29–7.41)	0.211

Mean [range]; * *p* value < 0.05, WBC Whole blood count; Hb Hemoglobin; PLT Platelet; CRP C reactive protein.

## Data Availability

The datasets generated and/or analyzed during the current study are available from the corresponding author on reasonable request.

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
