# Peer review of "Clinical and Laboratory Findings of Nosocomial Sepsis in Extremely Low Birth Weight Infants According to Causative Organisms"

_jcm, 2022, doi:10.3390/jcm11010260_

Round 1
Reviewer 1 Report
Thank you for allowing me to review this manuscript. This is a comparison of lab markers in three infectious category groups. The authors found difference in lab markers between Gram positives, negatives and fungal agents.
The methods are fine, and the results are accurate. I am concerned that the sample size is very small and I think requires much larger sample in each group.
The main concern is the conclusion, of using those marker to target therapy. In my opinion this is a far reaching conclusion that should be carefully considered. To use them as predictive marker, a different mehod and larger sample should be chosen.
Reviewer 2 Report
The article by Park et al. describes a cohort of newborns with extremely low birth weight that attracted infection during their admission at the NICU. The authors present general laboratory measures during the time of infection and analyze the test characteristics of the children comparing the children with a first gram positive infection with children with a first gram negative or fungal infection. The article is well written and discusses a difficult question related to choosing the right empirical therapy in case of suspected infection. The conclusion of the authors is that these general and routine laboratory measures could help in guiding the right empirical therapy.
The study is a classical retrospective study. The children are selected based on their outcome, i.e. a first infection with either a gram positive, gram negative or fungal infection. Children without an infection or culture negative infection were excluded. With the exclusion of the first group the test characteristics become very difficult, if not impossible to interpret. At start of symptoms it is unknown if a baby has an infection. This means that at the start of therapy it is unknown whether a patient fits into the selection criteria of this study. And children with a mildly elevated CRP changes are highest that there is no infection. But the authors conclude that this most likely fits with gram negative infection. But this is only the case if the blood culture becomes positive, after which the CRP hardly has any additional role in the diagnosis, but rather in follow-up. In addition the PPV of all the tests are rather low. How could this guide treatment decisions as suggested by the authors?
Another short coming is the lack of report of when all the laboratory measure are taken. The test characteristics of the laboratory values in infection are dependent on the timing since start of symptoms or start of antibiotics. Most reports show little discriminatory value at start, but this increases over time since start of symptoms and antibiotics. This is not reported in the manuscript. In addition in the discussion section the authors describe other reports looking at test characteristics mixing reports of lab tests at the time of starting antibiotics, with nadir values, without properly acknowledging these differences.
There is a high proportion of fungal infections in this cohort. Is fungal prophylaxis considered in these or future patients?
The wording late onset sepsis can be rather confusing, since term born children presenting at A&E until 3 months of age are also suspected of having late onset sepsis, but with a different presentation, and possible causing micro-organisms.
Round 2
Reviewer 2 Report
Again the design of the study does not fit with the conclusion. The authors excluded children without an infection or possible infection. The last group I understand, but the children with low CRP and negative culture are to be included if the data is to be used in practice. At the start of antibiotics the culture result is not known, so the clinician cannot judge whether the exclusion criteria apply to their specific patient. So the external validity is absent. The authors also do not mention what they mean with : These data may be helpful for choosing empirical antibiotics when sepsis is suspected. How? Because in most cases the culture remains negative and this study does not apply to that patient, but you’ll only know after 2 days. The design and the conclusion do not fit.
In the various outcomes mortality is not mentioned.
In the reply the authors mention the blood is taken at start. This should be clearly stated in the methods.
A better wording for sepsis in children at a NICU is nosocomial infection, rather than Late Onset Sepsis, since this term is also used for children that are not admitted to the NICU. The authors acknowledge this, but do not make any changements.
